# Oppositional Mirror on the Wall: Discursive Practices of Humorous *Pashkevilim* in Israel's Ultra-Orthodox Community

Hananel Rosenberg [1], Hila Lowenstein-Barkai [1] and Kalia Vogelman-Natan [2,*]

1 School of Communication, Ariel University, Ariel 40700, Israel; hananelro@gmail.com (H.R.); hilalow@gmail.com (H.L.-B.)
2 Media, Technology, and Society Program, Northwestern University, 2240 Campus Drive, Evanston, IL 60208-0001, USA
* Correspondence: kalia@u.northwestern.edu

**Abstract:** *Pashkevilim*, printed wall notices posted around Jewish ultra-Orthodox neighborhoods, serve as one of the religious community's popular communication channels. The *Pashkevilim* mostly deal with controversial intra-community issues and feature a unique style, extremist rhetoric, and vocabulary derived from the religious literature. Humorous imitations of the genre arose over the years, which circulated in the community and outside of it, posing a challenge to the rabbinic hegemony. Although humorous *Pashkevilim* have likely been present for as long as *Pashkevilim* themselves, there is currently a lack of research investigating them. By adopting a critical discourse analysis approach, the current study aims to address this gap by identifying the predominant types of humorous *Pashkevilim* and analyzing the discursive practices they employ. The findings indicate three main discursive practices that characterize humorous *Pashkevilim*: parody, satire, and irony. While parody exaggerates the formal characteristics of the genre and mocks them, satire and irony criticize the content and topics discussed in traditional *Pashkevilim*, especially on the subject of Jewish law and religious stringency. These practices express an oppositional reading of the genre, which challenges its function as well as its socio-cultural, political, and religious significance.

**Keywords:** *Pashkevilim*; humor; parody; satire; irony; Jewish ultra-Orthodox; Haredi Media





## 1. Introduction

The rapid advancement and adoption of digital technologies have reshaped communication through media, leading traditional media to seemingly become obsolete. While "old" forms of media are indeed not as strong as they once were, they have not completely disappeared, and in some specific contexts they still occupy a dominant space within the media landscape. One such noteworthy case is the printed wall notice known as the "*Pashkevil*" (pl. *Pashkevilim*); a centuries-old, non-official, public communication channel unique to the Jewish ultra-Orthodox (aka Haredi) community. Often written in Hebrew or Yiddish, the *Pashkevilim* are an inseparable part of the urban landscape of Haredi neighborhoods in Israel. In Haredi society, where mass media are subject to strict supervision and censorship, the *Pashkevilim* allow anyone from within the community to send a message to its larger target audience (Rosenberg and Rashi 2015). The generally anonymously posted *Pashkevilim* are used—among other things—to express ideological positions, expressing intra-sectoral critique, and the defamation of persons, phenomena, and competing world views.

Over the years, humorous imitations of the genre arose, serving entertainment-parodic purposes, satirical intents, and commercial objectives, etc. However, by being publicly circulated within and outside of the community, these humorous *Pashkevilim* pose a challenge to the rabbinic hegemony. Although humorous *Pashkevilim* have likely been present for as long as *Pashkevilim* themselves (i.e., since the advent of the printing press), there is currently a lack of research investigating them. The current study aims to address this gap in the

literature by identifying the predominant types of humorous *Pashkevilim* and analyzing the discursive practices they employ. By adopting a critical discourse analysis approach, the study offers insights into the socio-political role of humorous *Pashkevilim* within Haredi society and into the dialectical relationship between these particular texts and their cultural context.

## 2. Background Literature

### 2.1. The Ultra-Orthodox (Haredi) Community

Accounting for 13% of Israel's population, the ultra-Orthodox Jews, or *Haredim* (lit. "those who fear"), constitute one of the largest minority groups in the country (Cahaner and Malach 2021). Haredim are considered the most stringently religious Jews (Pinchas-Mizrachi et al. 2021) and are distinguished from other Jewish streams by several characteristics, including isolation from most influences of modernity and rejecting Western ideologies, sensibilities, and practices (Friedman 1991). In Haredi society, the constant study of Torah, both for its intrinsic value and in practical application, is considered the highest ideal (Cohen et al. 2021).

The Haredi community encompasses three primary sub-groups, each of which is further subdivided into distinct communities: Hassidim, Lithuanians (also referred to as *Mitnagdim*), and the Sephardic Haredim from mostly Middle Eastern and North African countries. Despite the differences amongst these three clusters, they all share a common characteristic of being governed by an authoritative leadership that exerts strict control over nearly every aspect of the group members' lives, including family life, education, and voting (Heilman 2000).

Power in the Haredi community is based on beliefs, dogmas, and taboos, resembling many authoritarian societies (Heilman 2000). Social control is used to enforce the key dogmas, but religious laws play an equally significant role in dictating behavioral standards within the community. Additionally, the community places a great deal of emphasis on rigid self-control and self-restriction, particularly in relation to traditional gender roles and extensive gender segregation (Stadler 2009). The community's stance on sexuality is highly conservative, limited essentially to the performance of marital duties, and forbids any spontaneous premarital interaction between men and women (Lev-On and Neriya-Ben Shahar 2011; Rosenberg et al. 2019).

The regulation of the group's media environment and the communicative practices of its members is another significant element of the authoritarian control. This control is exerted by the hierarchical leadership structure of the group, which oversees the strictly enforced self-discipline of community members (Sabag-Ben Porat et al. 2022). In Haredi society, mass media are subject to strict supervision. Since the Haredim isolate themselves from modernity, access to unfiltered internet, mobile phones, television, secular newspapers, and radio stations catering to the general public is prohibited (Neriya-Ben Shahar 2019). As an alternative, the Haredim have been most active in creating their own community media (Campbell and Golan 2011), which has traditionally taken on the role of preserving ultra-Orthodox community ideals by voicing rabbinical imperatives and reinforcing their hegemony (Laor and Galily 2022). The prevailing sectoral communication channels within the Haredi community include traditional media outlets like press and radio, as well as newer community-specific websites. Yet there is also a noteworthy, centuries-old communication channel that persists and maintains a significant role, known as the "*Pashkevil*" (pl. *Pashkevilim*).

### 2.2. The Pashkevil: Characteristics and Features

The *Pashkevil* poster, which is originally developed from wall notices, utilizes the low-cost and accessible distribution of printing. It has accompanied political, social, and religious conflicts in Europe since the 16th century. One of the *Pashkevil*'s important features is the capability of being distributed anonymously, which allowed its authors to challenge the leadership and the establishment (especially in Europe) without fear of condemnation,

ostracism, and punishment (Friedman 2005). This is particularly true among societies or sub-groups subjected to an authoritarian rule that prevents free criticism for religious or national reasons.

The *Pashkevilim* are an inseparable part of the urban landscape of Haredi neighborhoods. The *Pashkevilim* function in this society as a non-official communication channel used—among others—to express ideological positions, expressing intra-sectoral critique, and the defamation of persons, phenomena, and competing world views. Scholar Menachem Friedman (2005) explains that "the *Pashkevil* serves as a means for individuals and groups to mock and ridicule figures deemed authoritative and to challenge the ruling leadership... and is used by those who wish to protest against injustice and corruption" (p. 10). At times, the *Pashkevil* functions as a means for conveying vital intra-sectoral information and as a central source for news items, such as notices concerning gatherings and demonstrations, and publicizing rabbis' positions ("*Da'at Torah*," lit. "Knowledge of Torah") on a variety of contemporary issues. In most cases, however, the *Pashkevilim* deal with "burning" social-religious issues concerning the Haredi street, such as education and politics, Haredi enlistment to the army, modesty, and more (Charlap 2010).

Many of the *Pashkevilim* are written and distributed by isolated internal Haredi groups, such as "*Shomrei Ha'Chomot*" (lit. Guardians of the Walls) and "*Ha'Edah Ha'Haredit*" (The Haredi Council of Jerusalem), or by individuals wishing to protest internal and external Haredi phenomena (Rosenberg and Rashi 2015). In this manner they function as an alternative channel—and at times even oppositional—to institutional Haredi media ruled by the "mainstream" Hassidic and Lithuanian groups and are carefully monitored by "*Gedolei Ha'Dor*" (lit. the generation's greatest) and its representatives (Friedman 2005). However, in other instances the *Pashkevilim* are used by the "mainstream" Haredi leadership as a supplemental communication channel to the official media for distributing religious and social campaigns. For example, there has been a wide use of *Pashkevilim* on top of the general recruitment of the media in the public campaign waged in recent years to replace smartphones by distributing "kosher cellphones" (Rosenberg and Rashi 2015).

The *Pashkevilim* are characterized by changing levels of anonymity (Friedman 2005; National Library of Israel 2020). Notable rabbis—such as the rabbis of "*Ha'Edah Ha'Haredit*," "*Gedolei Ha'Dor*," Hasidic rebbes and heads of yeshivas—have signed off on only a handful of *Pashkevilim*. However, as mentioned, most of the *Pashkevilim* are anonymous or boast semi-formal monikers, such as "The Committee for Saving the Youth," "The Stimulators," and "The Committee for the Protection of Israel's Sanctity" (National Library of Israel 2020). For the most part, these are the independent initiatives of activists. Moreover, even when it is done at the bidding of the rabbis and the community's official leaders—the authors are the ones who determine the content and tone (Friedman 2005; Shapira 2005). Other studies have focused on the linguistic and syntactic aspects of the *Pashkevil* and the extreme rhetorical discourse that constitutes as one of the genre's identifying features (Schlesinger 1992; Kantor and Muchnik 2004; Shapira 2005; Charlap 2010). Hebrew linguist Luba R. Charlap (2010), for instance, identifies a variety of rhetorical means used by the *Pashkevilim* authors to achieve their objective, such as contrast patterns, rhetorical questions, repeated phrases, and the multiplicity of punctuation and emphasis. She argues that the profusion of these occurrences expresses a rhetorical cleverness that characterizes the *Pashkevilim* genre but is not unique to it and is found in other persuasion genres. Then again, phenomena like the abundant use of acronyms and the imitation of textual formats common in religious Jewish culture are unique to this genre and express its function as a distinct intra-sectoral communication channel.

*2.3. Humorous Pashkevilim*

Over the years there has been a widespread phenomenon of humorous *Pashkevilim* that imitate the classic *Pashkevil* features in their manner of writing. The phenomenon has been covered by news media, print and e-magazines, as well as online forums (Nachshoni 2014; Galili 2009; Borochov n.d.), but to date no scholarly research has been done on

humorous *Pashkevilim*. The authors of these *Pashkevilim* are diverse: individuals within and outside of Haredi society, institutions, advertising agencies, corporations, and more. These replications serve, among other things, entertainment-parodic purposes, satirical intents, and commercial objectives (Galili 2009). Thus, for example, in many cases the *Pashkevil* imitation is done within the family sphere, such as a humorous *Pashkevil* that serves as a sort of greeting for some occasion. The *Pashkevil* and its rhetorical style serve as a platform for the author drafting the greeting, whose humorous imitation does not relate to the *Pashkevil* features themselves.

Humorous *Pashkevilim* are not confined to private settings, but are often circulated in the public sphere, posing a challenge to the rabbinic hegemony. Although humorous *Pashkevilim* have likely been present for as long as *Pashkevilim* themselves, there is currently a lack of research investigating them. The current study aims to address this gap in the literature by identifying the predominant types of humorous *Pashkevilim* and analyzing the discursive practices they employ. By adopting a critical discourse analysis (CDA) approach, in which language is perceived as both a product and productive force in the construction of human experience (Van Dijk 1993), we seek to illuminate the socio-political role of humorous *Pashkevilim* within Haredi society and to contribute to understanding the "dialectical relationship between a particular discursive event and the situation, institution, and social structure which frames it" (Wodak and Meyer 2015, p. 5).

## 3. Methodology

### 3.1. Data Collection

A collection of *Pashkevilim* was compiled for the purpose of the study. The corpus was obtained from three sources: (1) the National Library of Israel's (NLI) online database, which includes approximately 15,000 *Pashkevilim* scans donated by a private collector, who had spent two decades taking down *Pashkevilim* and saving them in his home (to learn more, see Lidman 2011; National Library of Israel n.d.). A random sample of 350 *Pashkevilim* was taken from the collection; (2) *Pashkevilim* from the Haredi news and forums portal "BeHadrei Haredim," which was established in 2002 and allowed for online intra-Haredi critical discussion for the first time (Rose 2007; David and Baden 2020; Rosenberg and Rashi 2015). A search was performed for the term "*Pashkevil*" and related phrases, resulting in 100 *Pashkevilim* that were photographed and posted by users in various forums on the site; (3) *Pashkevilim* were also collected in a field observation conducted during the year 2020. This observation involved physically scanning two central streets in two Haredi neighborhoods in Jerusalem: Givat Shaul and Geula. The observation included a visit once every two weeks for six months (a total of 13 visits) to these streets and photographing *Pashkevilim* pasted on notice boards and buildings. As part of the observations, 230 *Pashkevilim* were photographed.

The combined corpus from these three sources included 680 *Pashkevilim* on various subjects. Aside from a handful *Pashkevilim* that feature a Hebrew date (in accordance with the Jewish calendar), most of the *Pashkevilim* are not dated and can only be estimated to have hung at some point within the particular source's sampling period as stated above. Furthermore, there is no way to determine the extent of the production (i.e., how many copies were printed) or distribution (i.e., how many were hung and where) of the *Pashkevilim*. Thus, the only criteria determining the inclusion of a *Pahshkevil* in the sample is that it had physically hung on the walls of Haredi neighborhoods, as verified by the NLI or a photograph. Two research assistants, both communications students, were employed to identify the humorous *Pashkevilim* within the corpus. Following a period of training, the two assistants and one of the authors read all the *Pashkevilim* individually and filtered out the non-humorous ones. After comparing the results and reaching a high level of interrater reliability, 28 humorous *Pashkevilim* were identified for further analysis conducted by the three authors.

About 45% of the sampled humorous *Pashkevilim* came from the online "BeHadrei Haredim" database, with the remainder sourced equally from the NLI and field observation

databases. Many of the sampled *Pashkevilim* overlapped in the sources, often appearing in two (and at times three) of the databases. This both lends the sample credibility and resolves potential biases related to the *Pashkevilim* distribution across sources.

*3.2. Analysis*

The present study employs the CDA framework, which proposes that the examination and interpretation of rhetorical strategies in a given text can elucidate the relationship between the text and its cultural context (Blommaert and Bulcaen 2000). CDA researchers view text as comprising three interrelated elements—namely content, structure, and language—which together provide insight into hidden political and social messages (Voloshinov and Bakhtin 1986; Kohn and Rosenberg 2013). The three authors, each with experience in studying religious groups in Israel and in conducting thematic (semantic and pragmatic) content analyses, all reviewed and analyzed the sample of humorous *Pashkevilim*.

In addition to analyzing the three CDA components, as well as in relation to previous studies that dealt with *Pashkevilim*, we focused on prominent rhetorical techniques, such as the presentation of the addresser and addressee, the nature of the reference, and key linguistic features (Charlap 2010). Furthermore, in our analysis we referred to the unique structural and rhetorical characteristics of humorous texts in their various forms, such as strategies of structural, linguistic, and textual incongruity (Palmer 1987), narrative contrast relations (Raskin 1985), as well as satirical (Rose 1993) and ironic (Sperber and Wilson 1981) hidden or overt meanings. The application of these parameters made it possible to identify three main types of discursive practices that characterized the humorous *Pashkevilim*: parody, satire, and irony.

## 4. Findings

*4.1. Parodic Pashkevilim*

### 4.1.1. Mocking the Genre Features

Parody and satire are similar in the dialogue that they have with an external cultural text and in the practice that ridicules the object they mimic. However, these genres differ from each other in the critical aspect of satire that is often lacking in parody, or in the very least not a significant part of it. For the most part, parody is limiting the mimicking, distortion, and radicalization of a certain text, using humor that mocks the original text's external characteristics, such as structure, rhetoric, genre features, and so on (Murfin and Ray 1997).

A classic example of a parodic *Pashkevil* can be seen in the following cited example (Figure 1). This reflective-*Pashkevil* mocks the morphological features of the classic *Pashkevil*, especially the *Pashkevil*'s unique font that characterizes the genre's traditional-minimalistic design, and emphasizes and highlights the expressions of crisis and outcry. The ironic aspect characteristic of these parodic texts (Shifman 2008) is prominent, among other things, in the technique of using expressions with high rhetorical force, such as "disaster!" and "cries of the land!" that are very noticeable on their own at first glance of the *Pashkevil*; however, when reading the entire *Pashkevil* and in combination with its parodic context, these words acquire a completely reversed meaning. For example, in the *Pashkevil*'s first line of "*Pashkevil* graphics simply look disastrous!" and in the middle section of the *Pashkevil*: "With all sorts of stupid phrases like cries of the land!" The incongruity of the "regular" appearance of the classic *Pashkevil* and the inverted and reflective appearance in this context create the parodic effect of the text.

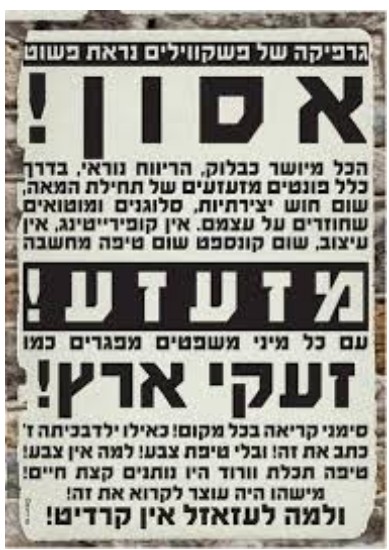

**Figure 1.** Cry of the fonts.

### 4.1.2. Incongruity and Content Reversal

One of the acceptable "incongruence" techniques in humoristic genres in general, and parody in particular, is the "mixing of fields" technique, in which the text combines elements from one cognitive, linguistic, or textual field together with elements from another field (Palmer 1987). Raskin (1985) analyzes these texts according to the "Script Principle," in which comic texts are characterized as consisting of two scripts that completely overlap and maintain a contrast relationship. An example of a parodic use of this technique is the humoristic *Pashkevil* that imitates a popular *Pashkevil* (which over time turned into a permanent billboard—Figures 2 and 3) warning neighborhood visitors from entering in attire that does not conform to the acceptable Haredi standard of modesty.

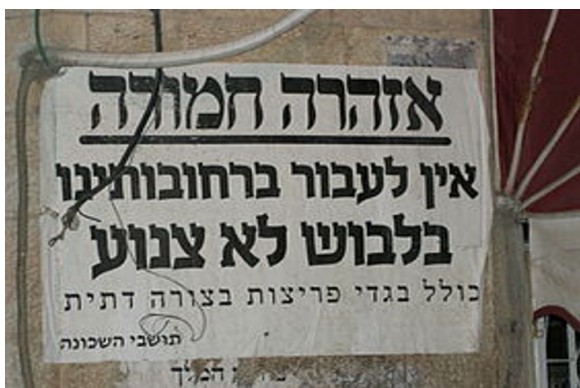

**Figure 2.** Warning *Pashkevil*. "Grave Warning: Don't pass through our streets in immodest clothes".

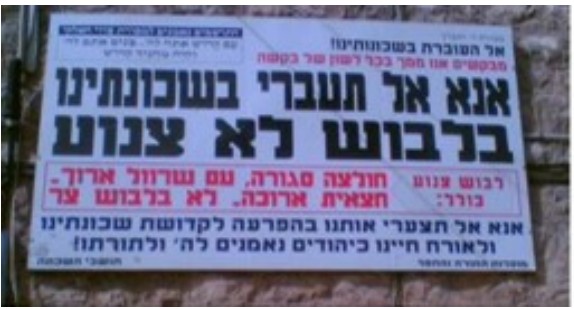

**Figure 3.** Warning Sign. "Don't pass through our neighborhood in immodest clothes".

The humoristic *Pashkevil* warns female visitors of Haredi neighborhoods from entering unless they are "dressed fabulously" (Figure 4). This *Pashkevil* deals with the rhetoric of the original text, and its humorous effect is a result of the incongruence in the encounter between the modesty discourse making use of religious rhetoric and the fashion discourse. The mixing of the fields also takes place in the very act of transforming the ban order into a fashion order, and in the phrases combining these fields within the same sentence, such as: "the holy brands," "with the help of the blessed brand," "the institutions of shopping and charity," and so on. The parody also exists through specific imitations of the original text, such as the modest attire list (e.g., "closed shirt with a long sleeve" in Figure 3) that turns into a detailed list of fashion items (e.g., "Dolce & Gabbana poncho, pink details in the collar," etc.,) in the humoristic *Pashkevil*.

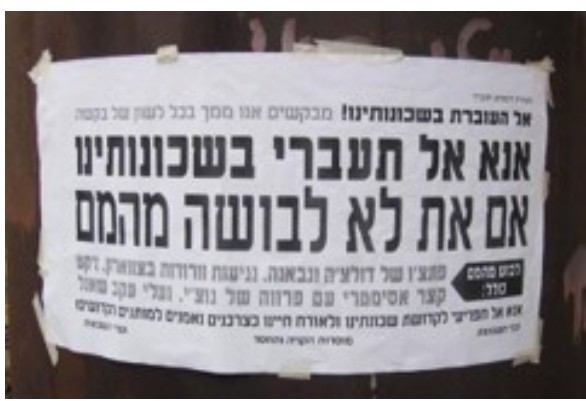

**Figure 4.** Parodic Warning *Pashkevil*.

*4.2. Satirical Pashkevilim*

4.2.1. Extremism as a Critical Practice

Contrary to parody, satire offers a critical view that is not limited to the imitation, distortion, and mockery of the genre and its characteristics. Satire uses humorous technique to criticize objects found outside the artistic work itself, such as deviations and violations of accepted social norms (Rose 1993). In this context, the reflective-satirical *Pashkevilim* do not use the *Pashkevil* imitation just to ridicule, for example, the visual or rhetorical characteristics of the genre, but rather to criticize the content and topics of the "regular" *Pashkevilim*. This is especially evident in humorous *Pashkevilim* on the subject of "halakhic stringency". The "stringency" (the addition of a prohibition or further adhering to the body of a commandment) is a common practice in religious-Haredi culture, and in a sense, is perceived as part of the self-definition of Haredi society, especially Ashkenazic (Liphshiz 2012).

The *Pashkevilim* have often been used to disseminate these "stringencies," such as strict observance of dietary laws, strict prohibitions concerning modesty, and other commandments, from baking matzah with "special elegance" to the purchasing of sukkah coverings under special halachic supervision. In this spirit, satirical *Pashkevilim* mimic the acceptable *Pashkevil* style, while inventing ridiculous stringencies. Thus, for example, there are those presenting Kosher-related dietary stringencies, such as the *Pashkevil* forbidding eating "Jerusalem Kugel" because the noodles look like worms (Figure 5).

Another example relates to the Haredi campaign against non-"Kosher cellphones," especially smartphones and their applications. The campaign led by the "Rabbinical Committee for Communications" was distributed in the official intra-sector press, as well as via the informal *Pashkevilim* channel (Rosenberg and Rashi 2015). These *Pashkevilim* described the danger of using ordinary smartphones in harsh terms: exposure to content services, "improper" Internet content, and inappropriate relationships between men and women that led to the wrecking of homes and families (Rosenberg and Blondheim 2021). In this context, a humorous *Pashkevil* that was published seeks to prohibit the use of

"Kosher cellphone" devices, claiming that some yeshiva students were addicted to the calculator found on these devices (Figure 6). This *Pashkevil* uses the official stamp of the Rabbinical Committee for Communications (a seal of Kosher certification imprinted also on the "Kosher cellphones" themselves), which exaggerates and ridicules the fear of addiction and lack of control by the user that led rabbis and activists to prohibit the use of many cellphone functions.

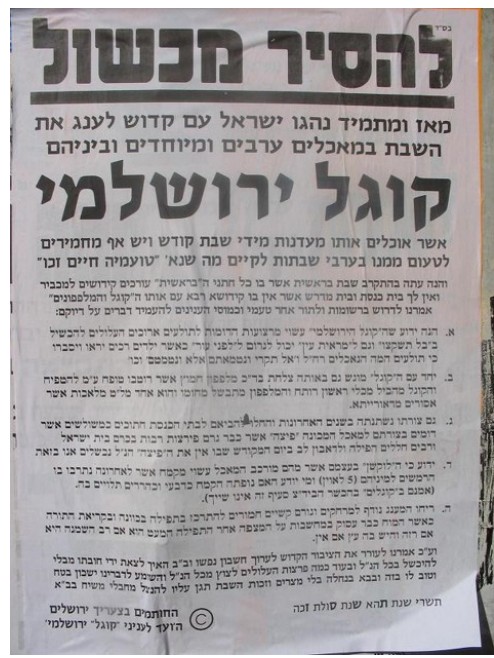

**Figure 5.** Kugel eating ban. "It is well-known that the 'Jerusalem Kugel' is made from strips that resemble long worms that can obstruct. . . when young children will see and discover that worms are eaten.".

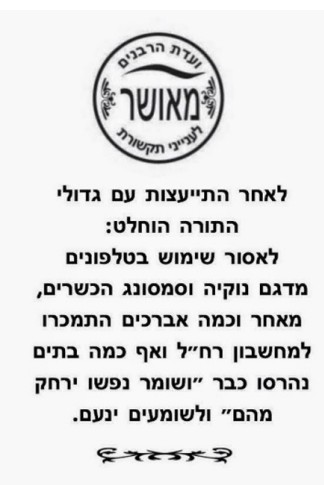

**Figure 6.** Calculator addiction.

The satirical *Pashkevil* is characterized by its reference to common phenomena that frequently appear in new "real" *Pashkevilim*. Thus, the issue of female modesty, which is broadly referenced in *Pashkevilim*, also receives a satirical *Pashkevil* calling to ban women from leaving their homes entirely (Figure 7). Another example is a satirical *Pashkevil* protesting Haredi organizations funded by Zionists. The real *Pashkevilim* on this subject are usually written by the "*Ha'Edah Ha'Haredit*" and the Haredi religious group of "*Neturei*

*Karta*" prohibiting cooperation with state institutions. This satirical *Pashkevil* (Figure 8) bans the use of Israeli money because it is "Zionist money". These examples, which exaggerate to the point of ridiculing the "discourse of prohibitions," seek to highlight the absurdity of the "ordinary" *Pashkevil*'s strict and extreme approach as well as its contents. In this sense, the satirical view of the *Pashkevil* complements the abovementioned parody effect in that both these humorous practices seek to expose the extreme characteristics of the *Pashkevil*, whether through its visual attributes and rhetorical style or in its radical content, and to undermine the basic infrastructures of the genre.

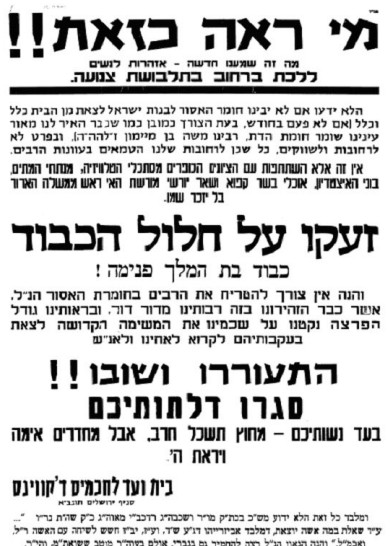

**Figure 7.** Banning women from leaving the home. *"Close your doors for your wives"*.

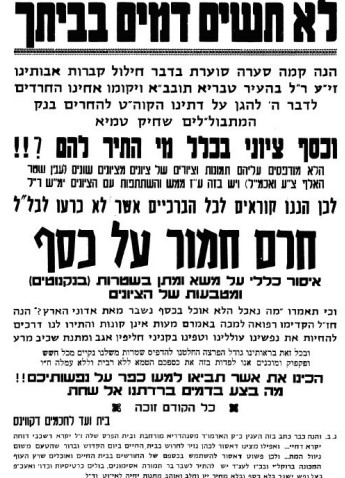

**Figure 8.** Boycotting "Zionist Money". *"General prohibition on negotiations with Zionists' banknotes and coins"*.

Another satirical style can be found in *Pashkevilim* that deal directly with a specific *Pashkevil*, as a kind of "focused response". For example, a *Pashkevil* using gematria[1] to claim that Internet equals cancer was distributed as part of a campaign dealing with the dangers of exposure to the Internet. The gematria, a commonly accepted rhetoric in the religious literature and Haredi discourse, is used to frighten and threaten Internet users (Figure 9). In response, similarly designed *Pashkevilim* using the same rhetorical style were distributed (Figure 10) to ridicule the rhetorical practice of gematria (satirically claiming that lettuce equals autism) and the threatening message of the original *Pashkevil*. However,

unlike the previous examples, in this case the author is not content with just the hidden satirical message, but rather exposes the critical message of the text in the final sentence at the bottom of the *Pashkevil* stating "There is faith and there is blindness".

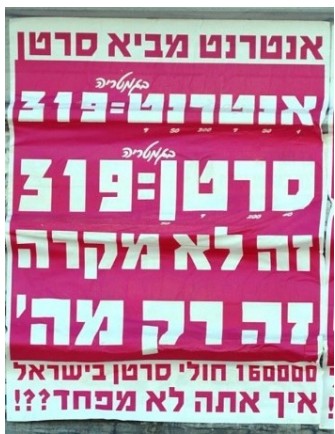

**Figure 9.** Warning (original).

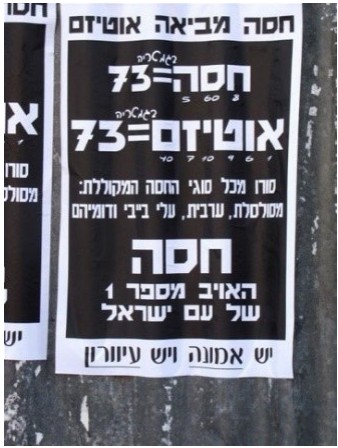

**Figure 10.** Warning (satirical).

### 4.2.2. Extremism and Counter-Extremism

Another example of a satirical *Pashkevil*, which was met with unexpected reactions, paradoxically demonstrating the very phenomenon the *Pashkevil* aimed to condemn, relates to the *Pashkevil* written by Prof. Moshe Kopel and Rabbi Daniel Guttenmacher (the former grew up in a Haredi household and is currently a mathematics professor, and the latter is a yeshiva rabbi). Both authors have a tradition of writing fake *Pashkevilim* every year in which absurd halakhic prohibitions are usually invented. They would have them printed and pasted across the streets of Haredi neighborhood Mea She'arim in Jerusalem (Galili 2009). In one instance, the authors anonymously distributed a *Pashkevil* dealing with matters of halakhic severity, in which they present a new severity that ostensibly prohibits drinking "non-Jewish water" because of the microscopic creatures found in it (Figure 11). The fake prohibition was quickly circulated and even made it into a CNN newscast as a real ban (Figure 12).

The cases in which satirical *Pashkevilim* were not recognized as such relate to a humor characteristic that requires radicalization in an attempt to create a humorous text, but not to over-radicalize the original text, so that the original text or genre will be understood. In the *Pashkevilim* genre we are dealing with, however, radicalization is a prominent feature of the original text itself. The *Pashkevilim* themselves make use of radicalization, in terms of both

the subjects and the rhetoric. Thus, sometimes also the over-radicalization of *Pashkevilim*, as perceived by the author, is not perceived as radicalization by the reader.

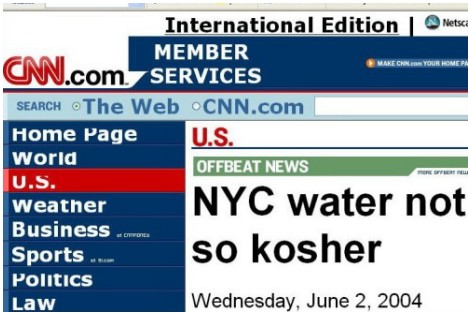

כי אם לשמוע את דברי ה'...

שמועה שמענו ותרגז נפשנו, הנה לאחר בדיקה יסודית במעבדות
של בית ועד לחכמים דקווינס נמצא שבכל המים שמספקים
הציונים הכופרים ימ"ש בין המים הנוטפים מן הברזים בין המים
הזוחלים מן המעיינות מסתתרות כמויות מחרידות של מזיקין
קטנים ר"ל הנראין לעיני הדיין דרך המכשירים המכונים
מיקרוסקאפען - אוי לעינים שכך רואות! - ודין אותם מזיקין
כדין שרץ המים ר"ל ולכן פשוט וברור שקיים

איסור חמור לשתות מים שלהם ח"ו

והשותה מימיהם עובר בששה לאוין מן התורה ר"ל והרי"ע, ואלה המקפידים לא
לשמור על חמר מדינה עאכו"כ שיתברכון ממי מדינה והמטמיר גם ברחיצת. פניו ידיו
ורגליו תבא עליו ברכה.
לכן בראותנו גודל הפירצה באנו אנו הנהלת חברת **מי מרה** מיסודם
של גדולי עסקנינו שליט"א שחזקו עליהם שחסים על ממונם וכו' לזכות
את הרבים למען ספות הרווה את הצמאה ולהתיר לציבור אנ"ש

לשתות וללוש במים שלנו בלבד

שהם על טהרת מי בורות שידין ומערות עוגורין (עי' חולין סו. ואבמ"ל), וכן מי מ"ט המלח הנקיים מכל חשש
זידקים וחיסוס פרטים ושאר שרצין זהב"ל בפשיחה בהשגחתו ובתסכמתו של ב"ק שה"ח שדראתו
קדמא לחכמנו והוכ כבוד גאב"ץ ק"ק סום החולת ת"ו הנד"צ **פ.צ. וואכטרטרעגער** ד"ע
בעצמהו"ס שמענו נא המורים וס' פזרה ראש ולענה.

הכל במחיר אנואה השוווה לכל נפש וללא פיקרון אף לפכים קטנים

כמו כן נזכה את הרבים במי חטאת מי גבאים מי רגלים מי סוטה ובסיט שהמים צפים
על גביהם ואי"ה בשעטו"מ ניפיק כמויות אדירות של מים כשרים בשיטת ההשקה
הידועה, ופשוט שדין מים קופצים המכונים קולא לחומרא ואלה. בטעם מענטא אסורים
אף מטעם ונשמרתם וכו' וד"ל ואין לנו אלא לשתות בצמא מדבריהם של גדולי רבותינו
**מבית ועד לחכמים דקווינס** שליט"א ואכמ"ל

**Figure 11.** A severe prohibition to drink their water. "We heard a rumor and our soul is angered [...] it has been discovered that all the water supplied by the Zionist heretics, may their names be obliterated [...] have horrifying amounts of small pests hidden within them".

**Figure 12.** CNN news coverage of the "ban".

*4.3. Ironic Pashkevilim and Trolling*
Impersonation as Subversive Practice

As seen above, in many cases the satirical or parodic text does not explicitly reveal that it is such, but rather expects the reader to identify its inherent criticism through its exaggerated and ridiculous style. However, in some cases the impersonation practice is used entirely in an attempt to convey the social criticism stemming from the reader's trust in the *Pashkevil*. In the spirit of Internet "trolling,"[2] this practice can be termed as "*Pashkevil* trolling," i.e., a humorous *Pashkevil* posing as a regular *Pashkevil* but in fact contains a subversive message intended for advancing a critical agenda.

An example of this practice was found during the 2014 Israeli elections when for the first time ever a political party made up of Haredi women was running for government. The party's slogan was "Not Elected, Not Voting," i.e., a protest against the Haredi norm that does not allow women to be elected to the Knesset (Israeli Parliament). As part of the political campaign, party members distributed an anonymous fictional *Pashkevil* protesting the party and mocking the women who wish to leave their traditional role in the home (Figure 13). This *Pashkevil* was intended to serve the reader as a mirror that reflects the status of women in the Haredi sector (Nachshoni 2014).

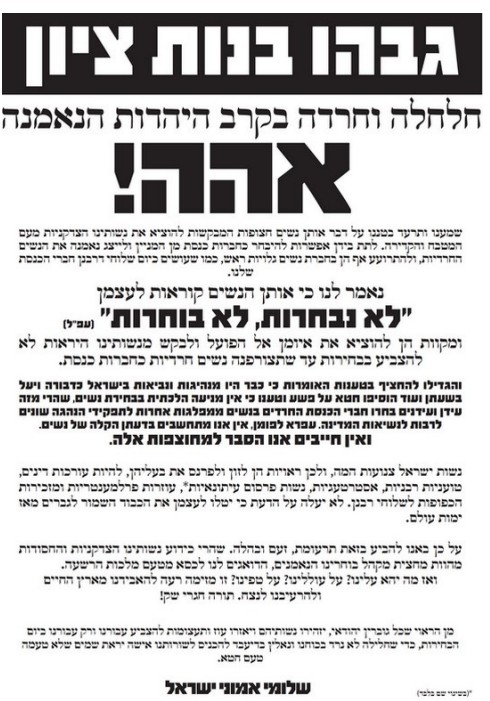

**Figure 13.** Not elected—not voting. "We heard of and tremble to our cores about those insolent women who wish to get our righteous women away from the kitchen and cauldron, to give them the opportunity to be elected as Knesset members and to socialize with bareheaded women [...] The women of Israel are modest and therefore worthy to sustain and support their husbands [...] it is conceivable that they will take upon themselves the honor reserved for men since ancient times".

The ironic aspect of this *Pashkevil* came into play in the gap between the distribution of the text on the Haredi street and the revelation of its true meaning. Irony is defined in Samuel Johnson's *A Dictionary of the English Language* as a "mode of speech in which the meaning is contrary to the words" (Johnson 1755; Sperber and Wilson 1981). Irony, by its definition, requires an understanding of opposites or of nuances in the meaning of expression that is contrary to the original statement. The recognition of an ironic text is fully realized only by identifying the comic element inherent in the reversal and the gap between the overt text and the hidden meaning behind it. In any case, the full meaning of the irony is formulated only after conflict and the creation of a gap between understanding the original meaning of the statement and the opposite meaning encoded in the specific context in which it was said. The contrast, or the multiplicity of colliding meanings, creates the desired "punch line".

Usually, the ironic decoding process takes place during the immediate encounter with the text. However, in the context of this *Pashkevil*, the inherent message is derived from the widening of the time gap between the absorption of the first, simple meaning of the text and the conflicting interpretation and understanding that this is an ironic text. Social criticism draws its strength precisely from this gap: the very ability of the addressee to believe that the text is real (and not ironic) for so long (until the media exposes that the *Pashkevil* is ironic) is a reflection of the bleak situation of women in Haredi society and realizes the inherent criticism of the text (Stadler and Taragin-Zeller 2017).

## 5. Discussion and Conclusions

In contrast to popular media texts that appeal to a wide range of audiences and are characterized in polysemy (Fiske 2010), the *Pashkevil* addresses a defined, local audience, and is usually characterized by the shutting out of others and one-dimensionality. However, it seems that the humorous *Pashkevilim* express a provocative and contradictory reading in its political sense (Hall 1993), which seeks to undermine the legitimacy of the original

text. Indeed, as Jensen (1990) points out, "resistive readings" often refer not only to a specific text but rather to the genre as a whole. Similarly, in the present context, the resistance that appears in the humorous texts is not only toward a focused text—such as a specific *Pashkevil*—but rather toward the entire genre, and in a sense, the genre's underlying attributes:

> The wider ramification of opposition on textual level depend on the social and political uses to which the opposition may be put in context beyond the relative privacy of media reception… this mean moving beyond the decoding of the individual text and focusing on the *genre*… its historical origins and designated social uses. (Jensen 1990, p. 58)

The reception of the humoristic Pashkevil as a real text is perceived by the author as a "victory". The author does not "celebrate" the *Pashkevil*'s success as an expression of protest, but rather more importantly that his protest was not understood as satire. His success frees the power of the absurd, as if the author says, "You are so extreme in your phenomena that you do not understand when it's serious and when it's a joke". This was the case with the Haredi women's election campaign (Figure 13), when the average Hardei audience believed it to be a real *Pashkevil* (Nachshoni 2014). Haredi websites' (such as "*Kikar HaShabbat*") reports on the *Pashkevil*'s reception also presented it as an authentic protest against the new political-feminine party, though all items were later removed when the truth about them being tricked was discovered (Nachshoni 2014). Indeed, the response to the *Pashkevil* proved the authors' point that this chauvinistic approach is perceived as logical in the Haredi sphere. This was also especially apparent in the case of over-radicalization of *Pashkevilim* (Figures 11 and 12). In the words of Israeli novelist Haim Be'er in response to the abovementioned water ban humorous *Pashkevil-inspired* confusion: "At the door of every satire lies the threat of reality" (Galili 2009).

Researchers point out that in many cases where satire makes use of parody, and these two genres are intertwined, it is possible to establish a polysemic text that is characterized by criticism against the object on the one hand, and by attraction and affinity on the other hand (Hutcheon 1985; Rose 1993). Indeed, it seems that even in this context, the parodic, satirical, and ironic *Pashkevilim* reflect an affinity toward the genre, style, and internal conventions. At the same time, however, they undermine the structural–rhetorical, content-related, and social infrastructure of the traditional channel on several levels.

First, the parodic *Pashkevil* that focuses on rhetorical infrastructure acts in the spirit of the formalist school of thought who view parody as a tool for exposing the structural basis of a literary genre (Shifman 2008). It can be argued that the parodic *Pashkevilim* function similarly, by underscoring and exposing the minimalistic external characteristics on the one hand, and the extreme rhetoric on the other. The satirical *Pashkevil*, however, deals with critiquing the religious–social content discussed in this genre, while radicalizing it and making it absurd.

It is possible that the parodic practices found in the current study, which exaggerate the formal characteristics of the genre and mock them, reflect the infiltration of new technologies into the Haredi community. Findings from recent years indicate that while religious leaders' ban of television gained large-scale compliance, restrictions on smartphone and Internet use proved less effective, with a penetration rate of 64% in 2020 (Cahaner and Malach 2021; Mishol-Shauli and Golan 2019). These studies suggest that the Internet offers a new religious and spiritual environment, providing a valuable prism for observation of numerous processes in the networked society (Okun and Nimrod 2017). In this state of affairs, *Pashkevilim* seem to be a particularly archaic communication channel, and their distinct style is an object of ridicule and criticism.

On another level, the ironic and "imposter" *Pashkevilim* focus on the audience and the reading experience, criticizing the way in which the audience accepts the genre and its content without question. It seems that the various humorous *Pashkevilim* genres expose the internal structures and conventions of the genre and its relations with the readers. The specific and unique social context of the *Pashkevil* clearly demonstrates Giddens's (1984)

structural approach, which claims that genre is both a medium and the result of textual practices defined in specific social environments: "Genres are both medium and outcome of textual practices in socially organized settings (social institutions, social fields, social systems, etc.)" (p. 25).

The individual's choice and the group's decision to use a specific genre is also the result of a social structure, but also a local creative and replicating practice (Berkenkotter and Huckin 1993). The literary or media genre is not neutral but rather has a defined set of expectations and hypotheses (Neale 1990). In this sense, any definition and use of a media genre is an act of marking and an act of inclusion and exclusion. This approach is especially relevant to the specific case of the *Pashkevil*, since discourse practices, rhetorical characteristics, and sectoral content create a basic alienation of the genre against external readers. It can also be said that in many cases for foreign readers the basic approach to this genre will be parodic, even toward ordinary texts. However, it turns out that even in the inter-community context, the structural and content-based extremism in a sense make the *Pashkevil* reflective in essence, and serve as a platform for imitation, alienation, and ridicule.

*Pleasure, Reflection, and Criticism*

Criticism is one of the main virtues of humor (Douglas 1975). The moment of exposing the humorous artifice and its accompanying pleasure is the reflective, critical moment in which the reader's readiness for self-disclosure and critical observation opens (Kohn 2013). This mental stance (of the addressee) enables the author to offer their critique: of the genre, of the social use of this channel, and of the manner in which the reader does not exercise proper criticism. This is one of the advantages of the humorous expression—parodic, satirical, or ironic—because social criticism, however poignant, comes from a non-threatening position and meets the recipient in the middle.

The reader's immediate understanding of the comic element in the text depends on the understanding of the discourse, the context, and the nature of the audience to which the message is directed. One of the basic practices of humorous expression lies in the ability to create a separate and sophisticated community of equals, whose members alone can fully understand the humor strategies and the nuances of irony. However, the clash between the ambiguities and challenges that this dialect represents not only addresses the *Pashkevil* reader as an individual, but resonates with the entire genre, its style, its social context, its religious significance, and its role as a traditional genre representing a conservative society. This meaning is especially relevant to the audience of *Pashkevil* readers, the Haredim, who are characterized by total passivity. In this sense, the satirical *Pashkevil* is a kind of critical dialogue (Elkins 2002) and a sort of response by the silent audience to the character and content of the one-way communication taking place on this channel.

A well-known Haredi proverb says that "the *Pashkevilim* are not only glued to the neighborhood walls, but rather hold them together" (Friedman 2005). As a paraphrase, it can be said that the humorous *Pashkevilim* seek not only to undermine the *Pashkevilim* that are hung on the neighborhood walls, but also to undermine some of the walls on which the *Pashkevilim* are hung.

**Author Contributions:** Conceptualization, H.R., H.L.-B. and K.V.-N.; Methodology, H.R. and H.L.-B.; Formal analysis, H.R., H.L.-B. and K.V.-N.; Writing—original draft, H.R., H.L.-B. and K.V.-N.; Writing—review & editing, K.V.-N. All authors have read and agreed to the published version of the manuscript.

**Funding:** This research received no external funding.

**Data Availability Statement:** The data presented in this study are available on request from the corresponding author.

**Conflicts of Interest:** The authors declare no conflict of interest.

**Notes**

1    A system of alphanumeric code in Jewish culture.

2    "Trolling" is Internet slang for when a person posts or comments online in a deliberate attempt to instigate arguments, conflicts, hostility, etc., taking pleasure in "baiting" and provoking people online.

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
