# Peer review of "Oppositional Mirror on the Wall: Discursive Practices of Humorous Pashkevilim in Israel’s Ultra-Orthodox Community"

_religions, doi:10.3390/rel14060717_

Round 1

Reviewer 1 Report

To the article author:

Review of "Oppositional Mirror on the Wall: Discursive Practices of Humorous Pashkevilim in Israel’s Ultra-Orthodox Community"

Thank you for the opportunity to review this fascinating manuscript concerning Haredi Pashkevilim. While the article deals with an interesting topic that is relevant to the journal, it is my opinion that minor revisions could be made to perfect the article in preparation for publication. I offer the following recommendations and questions with the hope that they will be beneficial as you move forward with your manuscript. I wish to relay that I enjoyed reading this insightful article, and I look forward to its future publication.

Literature review

1.      Please add sources to all claims and notions (e.g., 55-61; 108-116; 120-127; 140-152)

Methods

2.      Could you please add the sources of the 28 chosen documents regarding the three databases?

3.      Additionally, please contextualize any biases or meanings this distribution (from the three databases) could have on the findings. For example, maybe most of the documents are new and only from the visits, and this perhaps could posit that the phenomenon is a contemporary one (or that the different themes exist only in contemporary documents). Perhaps the final sample could have been weighted among the three databases. And if not, perhaps this should be explained or contextualized.

4.      Could you please write about the authenticity of Pashkevilim on the website in the second database – that they indeed appeared at some time on the streets of ultra-Orthodox neighborhoods.  

5.      Indeed, could you speak about the criteria of a Pahshkevil in the sampling? For example, were all Pashkevilim printed? And were they all glued to neighborhood walls? Did they have to be mass-produced, or was it enough if even one was put up? Etc. (For instance, was Figure 11 actually printed and glued in Haredi neighborhoods? Or was it just a joke between two friends?). Please explain this criterion and contextualize its meanings for the findings.       

6.      Could you please elaborate on the analysis process: for example, who conducted the analysis (the authors? The assistants? Others? ), and as the understanding of Pashkevilim is nuanced (as described in 471-474), what prior experience or knowledge did the person/s have in approaching this analysis.

Findings

7.      I suggest moving the section about "reception" (347-353) to the discussion, as the findings relate specifically to the content analysis of the documents and not to the audience.

8.      363-373. Please refrain from presenting the reader's reactions (audience) as this fall outside the empirical scope of the study and the sampled data and cannot be deduced from the content analysis. I suggest the authors rewrite this section without referring to the audience or move it to the discussion (and add sources to their claims about the reception, for example: "the fact that the average reader believed", "websites were tricked and presented", etc.).

9.      398. Please add a source.  

Reviewer 2 Report

I enjoyed reading this article. This is cutting-edge research, well-written, and excellent scholarship. I have no comment to the author—a job well done. It's also nice to have some laughs while reading an academic article. 
